Study on SARS-CoV-2 infection in middle-aged and elderly population infected with hepatitis virus: a cohort study in a rural area of northeast China

Pan Yuchen 1 2 3
Jia Zhifang 1
Yu Xinyi 1
Lv Haiyong 1
Zhang Yangyu 1
Wu Yanhua 1
Jiang Jing jiangjing19702000@jlu.edu.cn 1 2 3
1 Department of Clinical Epidemiology, the First Hospital of Jilin University , Changchun , China
2 Center of Infectious Diseases and Pathogen Biology, the First Hospital of Jilin University , Changchun , China
3 Department of Epidemiology and Biostatistics, School of Public Health Jilin University , Changchun , China
Upadhyay Rohit
Electronic publication date: 2025 Feb 21
Publication date: 2025
Volume: 13
Electronic Location ID: e19021
Received 2024 Sep 20; Accepted 2025 Jan 28
Copyright: ©2025 Pan et al.
Copyright year: 2025
Copyright holder: Pan et al.
License: This is an open access article distributed under the terms of the Creative Commons Attribution License, which permits unrestricted use, distribution, reproduction and adaptation in any medium and for any purpose provided that it is properly attributed. For attribution, the original author(s), title, publication source (PeerJ) and either DOI or URL of the article must be cited.
License URL: https://creativecommons.org/licenses/by/4.0/

Keywords: SARS-CoV-2, Hepatitis, Elder, Neutralizing antibody, IgG

Funding: National Natural Science Foundation of China No. 82373664 Scientific and Technological Development Program of Jilin Province No. 20200403098SF This study was supported by the National Natural Science Foundation of China (No. 82373664) and the Scientific and Technological Development Program of Jilin Province (No. 20200403098SF). The funders had no role in study design, data collection and analysis, decision to publish, or preparation of the manuscript.

==============================
Background

To investigate the symptoms, the level of antibody, the progression of liver disease after SARS-CoV-2 infection in middle-aged and elderly population infected with hepatitis virus.

Methods

The study was based on a cohort of high-risk liver cancer and the participants was recruited in April 2023. Blood sample were collected and information was obtained through questionnaires. Data on reinfection was obtained by follow-up until July 31, 2023. The SARS-CoV-2-specific neutralizing antibody and IgG were measured.

Results

A total of 599 participants infected with hepatitis virus were included and the mean age was 61.3 ± 7.4 years. The SARS-CoV-2 infection rate was 94.7%. Among the infected, 132 were asymptomatic, 435 were symptomatic, no severe cases occurred. Four months after infection, no difference was in liver function and aMAP score between the infected and uninfected. The infected had higher seropositivity rates of both antibodies than the uninfected (neutralizing antibody: uninfected: 93.7%, infected: 99.6%; IgG: uninfected: 59.4%, infected: 98.9%). The levels of both antibodies in the symptomatic were higher than those the asymptomatic and the uninfected (neutralizing antibody: uninfected: 0.75 AU/mL, asymptomatic: 15.46 AU/mL, symptomatic: 24.76 AU/mL; IgG: uninfected: 15.10 AU/mL, asymptomatic: 263.84 AU/mL, symptomatic: 291.83 AU/mL). By July 31, 2023, the incidence of reinfection was 17.5%.

Conclusions

Although the infection rate of SARS-CoV-2 was high, no severe cases occurred. Omicron infection may not aggravate progression of hepatitis. Four months after infection, the population showed high positivity rate in neutralizing antibody and IgG. Monitoring of virus mutations and targeted prevention and care strategies is crucial for vulnerable populations.

Introduction

Severe acute respiratory syndrome coronavirus 2 (SARS-CoV-2), initially identified in December 2019, is an RNA virus. The global disruption caused by its transmission resulted in the COVID-19 pandemic, which has led to more than 6,800,000 deaths worldwide, predominantly affecting the elderly and individuals with underlying health conditions (Biswas et al., 2020; Johns Hopkins Coronavirus Resource Center, 2022; Lazarus et al., 2022).

Due to weaker immune system, elderly individuals are susceptible to infectious diseases. China has the largest elderly population globally, with 46.0% of this population residing in rural areas (Chen et al., 2022; National Commission on Aging of China, 2024). The higher rates of SARS-CoV-2 infections and fatalities in rural areas can be attributed to factors such as limited access to healthcare, poverty, and insufficient dissemination of health information (Liu et al., 2023). Thus, it is crucial to prioritize timely attention to the rural elderly, who are particularly vulnerable to both contracting the virus and experiencing severe outcomes.

Recent studies have shown that SARS-CoV-2 not only affects the respiratory system but can also lead to liver injury (Huang et al., 2020). Approximately 50% of individuals with COVID-19 experience abnormalities in liver enzymes (Wang et al., 2020). Data from real-world cases indicate that the virus can exacerbate pre-existing liver conditions (Imam et al., 2023). Patients with chronic liver diseases are considered to be at high risk for severe COVID-19 infection (Toutoudaki & Androutsakos, 2024). Information gathered from registries such as COVID-HEP and SECURE-Cirrhosis further emphasize the impact of underlying liver diseases on the outcomes of COVID-19 (Kushner & Cafardi, 2020; Marjot et al., 2021). This evidence confirms that attention should be paid to the situation of individuals with liver disease after infected with SARS-CoV-2 (Zhang, Huang & Zhang, 2020).

Immunity mediated by antibodies triggered by natural infection or vaccination is essential for managing the COVID-19 pandemic (Wang et al., 2023). Human antibodies belonging to IgG, IgA, and IgM classes are essential players in COVID-19 immune response. Among antibodies produced against SARS-CoV-2, those that block the binding of SARS-CoV-2 spike receptor-binding domain (RBD) to its angiotensin-converting enzyme 2 (ACE2) receptor are called anti-S antibodies, (also known as neutralizing antibodies); they can reduce SARS-CoV-2 virulence (Duś-Ilnicka et al., 2023). In the immune response mounted against SARS-CoV-2 vaccination or infection, IgG antibodies are typically produced last, but are maintained the longest; they are believed to provide long-term protection (Scourfield et al., 2021). Previous studies have observed variations in antibody level between asymptomatic and symptomatic patients, suggesting that symptomatic infections may elicit distinct immune responses to the virus compared to asymptomatic individuals (Lei et al., 2021; Wellinghausen et al., 2020). However, it is not very clear yet.

On December 7, 2022, the Chinese government issued the “Ten New Measures” for handling SARS-CoV-2 infections. Furthermore, COVID-19 has been classified under Category B management in China following a comprehensive assessment. Since then, a significant SARS-CoV-2 Omicron variant epidemic has emerged in China after the easing of prevention and control measures (Jing et al., 2023). Here, we investigated a high-risk cohort for liver cancer following the relaxation of measures after mass vaccination in rural China. A cohort of middle-aged and elderly individuals infected with the hepatitis virus in rural areas of Fuyu city, Jilin Province, was investigated for symptoms of SARS-CoV-2 infection. Furthermore, the SARS-CoV-2-specific neutralizing antibody and IgG level among uninfected, asymptomatic, and symptomatic individuals were analyzed, and the impact of antibody level on reinfection was discussed. Simultaneously, the impact of SARS-CoV-2 infection on the progression of pre-existing hepatitis virus infection was also studied.

Materials & Methods

Study design and participants

The investigation was based on a high-risk cohort for liver cancer in the rural area of Fuyu city, Jilin Province (He et al., 2023). The participants were recruited between April 1 and 15, 2023. The participants received the periodic examination and were asked to complete a questionnaire on liver disease and COVID-19. Based on the cohort, the inclusion criteria were: (1) age: ≥ 45 years old as of April 1, 2023; and (2) hepatitis virus infection: infected with hepatitis B virus (HBV), or previously or currently infected with hepatitis C virus (HCV): as evidenced by positivity for hepatitis B surface antigen (HBsAg) and/or hepatitis C antibody (HCV-Ab). A flowchart for participant enrollment was shown in Fig. 1. The participants’ information was collected and anonymized for analysis. Written informed consent was obtained from each participant. The study was approved by the Ethics Committee of the First Hospital of Jilin University (2023-455).

Data collection and follow-up

All eligible participants completed a standardized questionnaire. The questionnaire included information on liver disease treatment since the previous follow-up (2020) and SARS-CoV-2 infection status. Confirmed SARS-CoV-2 infection was determined through self-report with positive for SARS-CoV-2 on a rapid antigen test. The date of infection and symptoms during the infection were further collected, if the participant was infected. In addition, the questionnaire also covered the COVID-19 vaccination status and treatment measures. For the inactivated SARS-CoV-2 vaccine, receiving two doses was considered as completing the basic immunization schedule. Asymptomatic infection was defined as the absence of all of the following symptoms after infection: fever (>37.5 °C), chills, cough, sore throat, nasal congestion, headache, myalgia, fatigue, rhinorrhea, hyposmia, hypogeusia, dyspnea, sputum production, hemoptysis, dizziness, anorexia, nausea, vomiting, abdominal pain, and diarrhea. Symptomatic infection was characterized by the presence of any of the aforementioned symptoms after infection. Additionally, the questionnaire also covered the content of long-lasting symptoms of SARS-CoV-2 infection (“long COVID-19” symptoms). The definition of long COVID was as the presence of at least one symptom included in the WHO case definition at ≥12 weeks after infection (Subramanian et al., 2022).

Study participants were further interviewed via telephone to gather data on SARS-CoV-2 reinfection. Follow up was until July 31, 2023. Cases of SARS-CoV-2 reinfection, occurring > 60 days after the last infection, were identified through self-report (with a positive result for SARS-CoV-2 on a rapid antigen test). None of the participants in the cohort were reinfected with SARS-CoV-2 in April 2023, and all recorded reinfections occurred thereafter.

Sample collection and laboratory testing

After completing the questionnaire, five mL of venous blood from each participant was collected and stored in a −80 °C low-temperature freezer. The iFlash-2019-nCoV IgG assay commercial detection kit (Shenzhen YHLO Biotech Co. Ltd., Shenzhen, China) was utilized for quantifying IgG levels by chemiluminescence immunoassay. According to the protocol, titers of ≥10.0 AU/mL are considered positive (or reactive).

The neutralizing antibody (including IgA, IgG, and IgM) against SARS-CoV-2 RBD was measured by double-antigen ELISA kit (Hotgen Co. Ltd., Beijing, China) by upconversion luminescence immunochromatography. The positive cut-off value was defined as titers of ≥0.25 AU/mL. The upper limit for the detection was 99 AU/mL. In subsequent statistical analysis, any values exceeding the upper limit would be assigned the value of 99 AU/mL. The tests were conducted in accordance with the manufacturer’s guidelines.

HCV RNA load and liver function tests(alanine aminotransferase (ALT), aspartate aminotransferase (AST), γ-glutamyl transpeptidase (GGT), alkaline phosphatase (ALP), lactate dehydrogenase (LDH), total bilirubin (TBIL), direct bilirubin (DBIL)) were performed by Jilin KingMed Clinical Laboratory Co., Ltd (Changchun, China). The testing methods are detailed in Table S1.

Statistical analysis

Continuous variables were described as either mean ± standard deviation for normally distributed data or as median with interquartile range (IQR) for non-normally distributed data, and differences among groups were assessed by analysis of variance or Kruskal–Wallis H test, respectively. Categorical variables were displayed as frequencies with percentages and were analyzed using χ2 test or Fisher’s exact test. In multiple pairwise comparisons between groups, the significance threshold was adjusted using Bonferroni’s method. Data on liver function from previous follow-up (2020) were used as baseline for analysis (at that time, no participants were infected with SARS-CoV-2). The age-male albumin-bilirubin-platelets score (aMAP score) was calculated for each patient at two visits as follows: aMAP score = ((age (year) × 0.06 + gender × 0.89 (male: 1, female: 0) + 0.48 × ((log10 bilirubin (µmol/L) × 0.66) + (albumin (g/L) × −0.085)) − 0.01 × platelet count (103/mm3)) + 7.4)/14.77 × 100 (Fan et al., 2023). The generalized Estimating Equations model (GEE) was used to analyze the impact of time and SARS-CoV-2 infection on liver function and the aMAP score. Spearman’s correlation coefficient was utilized to describe correlation between IgG and neutralizing antibodies. Linear regression analysis was conducted to assess the variables impacting the antibody level, and both antibodies data were ln transformed prior to the linear regression. Statistical analysis and plotting were performed using SPSS software (SPSS Inc., Chicago, IL, version 26.0) and GraphPad Prism 8 (San Diego CA, USA). A two-tailed P value < 0.05 was considered statistically significant for all tests.

Results

Characteristics of the cohort

As shown in Fig. 1, 599 participants were included. Between November 2022 and April 2023, 94.7% of participants were infected with SARS-CoV-2 for the first time, while the remaining participants had not been infected. As shown in Table 1, thirty-two (5.3%) reported no infection with SARS-CoV-2, 132 (22.0%) reported no symptoms after infection, and 435 (72.7%) reported at least one symptom after infection. The mean age of all participants was 61.3 ± 7.4 years. Participants without SARS-CoV-2 infection were older (mean age in years: uninfected: 67.3 ± 6.0; asymptomatic: 61.5 ± 6.9; symptomatic: 60.8 ± 7.4) and less likely to be male (male uninfected: 40.6%), compared to participants with asymptomatic infection (male asymptomatic: 73.5%) and symptomatic infection (male symptomatic: 60.2%). The participants in all the three groups were predominantly previously or currently infected with HCV, and the proportion of cirrhosis was highest in uninfected participants (25.0%) compared with the asymptomatic (17.4%) and the symptomatic participants (10.3%).

Figure 1 Flowchart for study participants.

Table 1 Characteristics of the participants.

Variable	All
n = 599	Uninfected
n = 32	Asymptomatic
n = 132	Symptomatic
n = 435	P	
Gender					<0.001	
Male	372 (62.1)	13 (40.6)	97 (73.5)	262 (60.2)		
Female	227 (37.9)	19 (59.4)	35 (26.5)	173 (39.8)		
Age (Y)	61.3 ± 7.4	67.3 ± 6.0	61.5 ± 6.9	60.8 ± 7.4	<0.001	
Hepatitis virus infection					0.055	
Only HCV infected	543 (90.7)	30 (93.8)	124 (93.9)	389 (89.4)		
Only HBV infected	41 (6.8)	1 (3.1)	3 (2.3)	37 (8.5)		
HCV & HBV co-infected	15 (2.5)	1 (3.1)	5 (3.8)	9 (2.1)		
Liver cirrhosis	76 (12.7)	8 (25.0)	23 (17.4)	45 (10.3)	0.010	
NO. of Vaccination received					<0.001	
≤1	11 (1.8)	4 (12.5)	2 (1.5)	5 (1.1)		
2	66 (11.0)	10 (31.3)	16 (12.1)	40 (9.2)		
3	522 (87.2)	18 (56.2)	114 (86.4)	390 (89.7)		
Duration from the last SARS-CoV2 vaccination to blood collection (Mo, (M,IQR))	15.6 (13.8–15.9)	15.7 (13.7–19.9)	15.6 (13.4–16.0)	15.6 (13.8–15.9)	0.588	
Duration from infection to blood collection (Mo, (M,IQR))	4.0 (4.0–4.0)	/	4.0 (4.0–4.0)	4.0 (4.0–4.0)	0.534	

SARS-CoV-2 vaccination status

Before blood collection, the overall vaccination rate for SARS-CoV-2 (at least two doses of the vaccine) was 98.2%. The majority of participants in all three groups were fully vaccinated and boosted with one vaccine dose (the uninfected: 56.2%, the asymptomatic: 86.4%, the symptomatic: 89.7%) or primed with two vaccine doses (the uninfected: 31.3%, the asymptomatic: 12.1%, the symptomatic: 9.2%). All vaccinated participants were primed with SARS-CoV-2 inactivated vaccines. The majority of the three groups were boosted with inactivated vaccines, some were boosted with recombinant protein vaccine pertaining to the recombinant SARS-CoV2 vaccine (CHO cell) (all: 5.7%, the uninfected: 5.3%, the asymptomatic: 6.7%, the symptomatic: 5.5%). The median duration from the last vaccine dose to blood collection was 15.7 months in the uninfected participants, 15.6 months in the asymptomatic and the symptomatic participants. The median duration from infection to blood collection was 4 months for both the symptomatic and the asymptomatic participants (Table 1).

Symptoms after infection with SARS-CoV-2

No serious cases occurred among the 435 symptomatic participants. More than half experienced fever (52.4%), followed by cough (41.4%); and about a third experienced myalgia/joint pain (29.6%). Other commonly reported symptoms included sore throat (14.9%), general malaise (14.3%) fatigue (13.8%) and headache (11.8%). The specific symptoms are depicted in Table S2.

Effect of SARS-CoV-2 infection on liver function

The effect of SARS-CoV-2 infection on liver function was evaluated using GEE model. After adjusting for age, gender, type of hepatitis virus infection, history of liver disease treatment, no significant differences were observed between the infected individuals and the uninfected individuals regarding ALT, AST, GGT, ALP, LDH, TBIL and DBIL (all P > 0.05) (Table S3). Nonetheless, a decrease in the proportion of participants beyond the normal range was observed in several indexes including ALT (OR = 0.60, 95% CI [0.39–0.92]), AST (OR = 0.51, 95% CI [0.37–0.71]) and LDH (OR = 0.32, 95% CI [0.24–0.44]) during this follow-up period compared to results from previous follow-up (2020).

The impact of SARS-CoV-2 infection on liver cancer risk, as assessed by the aMAP score, was also evaluated using the GEE model. The aMAP score was not significantly affected by infection (P > 0.05). However, it was noted that aMAP score was elevated at this follow-up compared to previous follow-up (2020) (aMAPpre-uninfected: 55 (49–59), aMAPpre-infected: 53 (49–57), aMAPpost-uninfected: 55 (51–61), aMAPpost-infected: 54 (50–58), Ppost-pre < 0.001), suggesting an increased risk of liver cancer with age (Table S4). Similar results were observed among participants with cirrhosis (Table S5).

SARS-CoV-2 serum antibodies in different groups

The median level of SARS-CoV-2-specific neutralizing antibody was 0.75 AU/mL, 15.46 AU/mL and 24.76 AU/mL for uninfected, asymptomatic, and symptomatic participants, respectively (Table 2). The neutralizing antibody level was significantly lower in uninfected participants compared to the infected (P < 0.001), and the antibody level in the symptomatic group was significantly higher than that in the asymptomatic group (P = 0.001). Based on the positive threshold cutoff value of neutralizing antibody, all three groups exhibited a positive rate of over 90% (uninfected: 93.7%, asymptomatic: 100%, symptomatic: 99.5%).

Table 2 Comparison of specific neutralizing antibody and IgG among three groups.

Variable	Uninfected
n = 32	Asymptomatic
n = 132	Symptomatic
n = 435	P	
Neutralizing antibody (AU/mL; M,IQR)	0.75 (0.44–1.31)	15.46 (5.06–36.06)	24.76 (8.27–80.51)	<0.001	
Neutralizing antibody classified				0.020	
<0.25 AU/mL	2 (6.3)	0	2 (0.5)		
≥0.25AU/mL	30 (93.7)	132 (100)	433 (99.5)		
IgG (AU/mL; M,IQR)	15.10 (4.37–36.17)	263.84 (157.12–325.49)	291.83 (222.07–348.65)	<0.001	
IgG classified				<0.001	
<10 AU/mL	13 (40.6)	1 (0.8)	5 (1.1)		
≥10 AU/mL	19 (59.4)	131 (99.2)	430 (98.9)		

The median level of SARS-CoV-2-specific IgG was 15.10 AU/mL, 263.84 AU/mL and 291.83 AU/mL for uninfected, asymptomatic, and symptomatic participants, respectively. Similar to neutralizing antibody, uninfected participants had lower IgG level compared to the infected participants (P < 0.001), and the IgG level of symptomatic participants was significantly higher than that of asymptomatic participants (P = 0.001). The seropositivity rate of IgG among uninfected participants was 59.4%, which was significantly lower than that among infected participants (P < 0.001). The seropositivity rate in the asymptomatic and symptomatic participants was similar (asymptomatic: 99.2%, symptomatic: 98.9%, P > 0.017).

A positive correlation was observed between IgG and neutralizing antibody level across all participants (rs = 0.723, P < 0.001), as well as within groups of different infection statuses (uninfected rs = 0.543, P = 0.001; asymptomatic rs = 0.712, P < 0.001; symptomatic rs = 0.667, P < 0.001), Fig. S1.

Factors related to antibody response

Linear regression analysis showed that gender may affect neutralizing antibody (β = 0.280, P = 0.019), but has no effect on IgG level (β = 0.034, P = 0.572); additionally, an increased number of administered doses corresponded to higher level of both antibodies (Neutralizing antibody: β = 0.642, P < 0.001; IgG: β = 0.422, P < 0.001). Compared to the uninfected, the asymptomatic and the symptomatic participants exhibited significantly higher levels of both antibodies (Neutralizing antibody: Asymptomatic-Uninfected β = 2.768, P < 0.001, Symptomatic-Uninfected: β = 3.114, P < 0.001; IgG: Asymptomatic-Uninfected: β = 2.733, P < 0.001, Symptomatic-Uninfected: β = 2.917, P < 0.001). The full regression model results were presented in Table 3.

Table 3 Multiple linear regression analysis of neutralizing antibody and IgG.

	Variable	β	P		Variable	β	P	
	Gender	0.280	0.019		Gender	0.034	0.572	
	Age	−0.002	0.847		Age	−0.001	0.746	
Neutralizing antibody	No. of vaccination	0.642	<0.001	IgG	No. of vaccination	0.422	<0.001	
	Group				Group			
	Uninfected	ref		Uninfected	ref	
	Asymptomatic	2.768	<0.001		Asymptomatic	2.733	<0.001	
	Symptomatic	3.114	<0.001		Symptomatic	2.917	<0.001	

Long COVID

A 26.1% prevalence of long COVID was observed. The most commonly reported symptoms was fatigue (11.8%), followed by cough (4.4%), myalgia/joint pain (3.4%), hypomnesia (2.1%), and dyspnea/asthma (2.1%). The proportions of other symptoms were much lower (Table S6). No significant difference was observed in the levels of both antibodies between participants with or without long-COVID (Neutralizing antibody: long COVID:25.40 (9.85–85.59) AU/mL vs. without long COVID: 20.30 (7.44–59.92) AU/mL, PNeutralizing antibody = 0.119; IgG: long-COVID: 287.12 (191.71–358.71) AU/mL vs. without long COVID: 288.14 (211.04–337.42) AU/mL, PIgG = 0.797. Table S7). No factors related to long COVID-19 were identified (Table S8).

SARS-CoV-2 reinfection

Participants infected with SARS-CoV-2 were followed until July 31, 2023. Of the 411 participants successfully followed up, 72 reported SARS-CoV-2 reinfection, yielding a reinfection incidence of 17.5%. The reinfection rate in the symptomatic group and asymptomatic group was similar (P > 0.05). The levels of the two antibodies were compared between participants who experienced reinfection and those who did not, and no statistically significant difference was observed (neutralizing antibody: reinfection:18.03 (7.57–49.02) AU/mL vs. without reinfection: 22.93 (8.02–62.04) AU/mL, Pneutralizing antibody = 0.477; IgG: reinfection: 268.62 (179.05–328.76) AU/mL vs. without reinfection: 286.49 (210.02–339.68) AU/mL, PIgG = 0.148. Table S7). No influencing factors related to reinfection were identified (Table S9).

Discussion

In the context of China’s transition from “dynamic zero-COVID policy” to “reopening after lockdown”, our study of a high-risk cohort comprising hepatitis virus-infected middle-aged and elderly individuals revealed that most participants experienced mild SARS-CoV-2 infection within the short period of time, with no observed progression of hepatitis. Higher level of neutralizing antibody and IgG was detected in COVID-19 patients, particularly in symptomatic cases. Also, antibody level didn’t impact reinfection.

In this study, between November 2022 and February 2023, 87.5% of the hepatitis virus-infected middle-aged and elderly participants were infected with the SARS-CoV-2 Omicron variant, and from November 2023 to April 2023, the infection rate was 94.7%, while the full vaccine coverage rate was 98.2%. According to the Chinese Center for Disease Control and Prevention, approximately 84.7% of the Chinese population had been infected with SARS-CoV-2 by February 2023 (Di et al., 2023). The study population exhibited a slightly higher infection rate compared to the general population in China. A mendelian randomization study also found that chronic viral hepatitis increased the susceptibility of the East Asian population to COVID-19 infection (Liu et al., 2023). Moreover, age may affect the prevalence of COVID-19, particularly among the elderly in rural areas who face an increased susceptibility to SARS-CoV-2 because of a lack of self-protection awareness and protective capabilities (Liu & Huang, 2023; Ren et al., 2022). The high prevalence underscores the necessity of targeted public health interventions for high-risk populations, such as those with chronic viral hepatitis and the elderly in rural areas. Public health education is also essential to enhance self-protection awareness and mitigate the spread of SARS-CoV-2. Further exploration of the links among age, comorbidities and virus susceptibility is warranted to facilitate better prevention and resource allocation.

In this population with high-risk factors, 23.3% remained asymptomatic following infection with the Omicron variant, whereas 76.7% exhibited symptoms. The COVID-19 symptoms observed, here, were mild, including fever, cough, myalgia/joint pain, and sore throat. Some individuals recovered at home or following outpatient treatment at clinics, and none required oxygen or mechanical ventilation. The main symptoms of Omicron variant infection in the general population are fever, cough, sore throat, and myalgia, as observed here (Sha et al., 2023). It is believed that the Omicron variant has less pathogenic potential than previous variants but possesses a greater ability to infect the upper respiratory tract (Bouzid et al., 2022). Individuals infected with the Omicron variant typically experience mild symptoms or may remain asymptomatic, due to the presence of a certain level of vaccine-induced immunity (Shao et al., 2023). Given these findings, public health could promote home-based care for mild cases to reduce healthcare strain.

Globally, the incidence of long COVID varies regionally. A survey conducted in the UK showed that the incidence of long COVID was 4.5% among omicron-infected individuals (N = 56,003) (Antonelli et al., 2022). During the omicron outbreak, a study conducted in China found that approximately 8.89% of individuals suffered from long COVID (Cai et al., 2023). Here, the prevalence of long COVID stood at 26.1%, with fatigue being identified as the predominant symptom. Fatigue also represents a prevalent symptom among individuals infected with hepatitis virus which might contribute to the high incidence of long COVID in this population.

Given that all subjects were infected with the hepatitis virus, liver function and liver cancer risk was analyzed in this study. The results showed that there was no discernible variance in liver function between SARS-CoV-2-infected and uninfected participants, approximately four months after the infection. Further, the aMAP score in uninfected and infected groups were similar indicating no difference in the risk of liver cancer between the two groups. Therefore, infection with the Omicron variant may not exacerbate liver disease progression in individuals infected with hepatitis virus. However, the molecular mechanisms underlying viral infections are highly complex, and these macroscopic observations may not directly reveal the causal relationships at the molecular level. In addition, it should be noted that, the aMAP score of the participants in the 2023 follow-up increased compared to the previous follow up (2020), suggesting a higher risk of liver cancer with age. Regular follow-up examinations remain essential for individuals at high risk of liver cancer.

The levels of neutralizing antibody and IgG specific to SARS-CoV-2 were also assessed four months post-infection. We found that the infected participants exhibited higher level of neutralizing antibody and IgG and higher seropositive rates compared to uninfected patients, after adjusting for the vaccine dose. Other studies have reported findings similar to ours, but in different populations. Following vaccination, participants who had been previously infected with SARS-CoV-2 had higher titers of specific antibody compared to those who had not been infected, as indicated by a study conducted in the United Kingdom (Manisty et al., 2021). A study by Abu Jabal et al. (2021) found that individuals who had been previously infected had antibody levels that were one magnitude higher than those who had not been infected before. It has been proposed that the natural infection of SARS-CoV-2 might elicit the varied polyepitopic cellular immune response that specifically directed at the nucleocapsid protein, spike protein, and membrane protein of the virus. Collectively, data in the current research, along with the results of prior studies, indicate that COVID-19 infection elicits a more comprehensive immune response. Moreover, it seems that there is a variation in the immune response in asymptomatic versus symptomatic individuals infected with SARS-CoV-2. Our data indicated the titers of neutralizing antibody and IgG were higher in symptomatic individuals compared to asymptomatic patients four months after infection, consistent with findings from previous research. A study in Beijing found six months after confirmation, patients showing symptoms had higher level of neutralizing antibodies compared to patients without symptoms (Cui et al., 2022). An study that also reported similar results: the asymptomatic individuals exhibited lower level of IgG compared to that in the symptomatic individuals (Shirin et al., 2020). These results imply that even when the symptoms are mild, asymptomatic patients had markedly lower serological response compared to symptomatic patients. Immune responses in asymptomatic infection primarily consist of innate, adaptive, and vaccine-induced immune responses. While adaptive immunity plays a significant role in asymptomatic cases, the features of T cells might exhibit variations compared to symptomatic infections (Boyton & Altmann, 2021). This may explain the variation in antibody levels between symptomatic and asymptomatic individuals, but further research is warranted.

Our results showed that the seropositivity of SARS-CoV-2-specific neutralizing antibody and IgG was extremely high four months post-infection. Antibody, particularly the neutralizing antibody is widely acknowledged as a critical indicator of immune protection after the infection is resolved (Kim et al., 2021). However, despite high seropositivity, 17.5% of participants still experienced a reinfection within 7 months after the initial infection, and no correlation between antibody level and reinfection. We suspect that this phenomenon may result from infection by different viral subvariants. The predominant SARS-CoV-2 strains in mainland China were Omicron BA.5.2.48 and BF.7.14 from December 2022 to the end of January 2023, while Omicron XBB was the main epidemic strain, from May 2023 to July 2023 (Feng et al., 2024). The XBB subvariant is considered as the most infectious subvariant, capable of evading neutralizing antibody (Imai et al., 2023). Clinical studies have shown notably higher rates of reinfection associated with XBB subvariant, compared to those associated with previously circulating variants (COVID-19 Forecasting Team, 2023; Özüdoğru, Bahçe & Acer, 2023). The high reinfection rates despite high seropositivity of antibody underscore the necessity for ongoing public health surveillance and the adaptation of vaccine strategies to target emerging subvariants like XBB, which have shown increased ability to evade existing immunity.

There are several limitations in our study. First, we confirmed COVID-19 cases by self-reporting a positive result for SARS-CoV-2 on a rapid antigen test, which may have led to some bias. But under the dynamic zeroing policy, local residents had used antigen test kits many times and were familiar with the use of the kit. Therefore, the potential bias is minimal. Second, during the follow-up, 27.5% of participants were lost to follow-up and data on reinfection among them was not available. However, the characteristics of participants lost to follow-up were similar to those who remained, indicating that the lost participants represented a random subset, and thus, minimally impacted the results. Third, the study was conducted in a specific geographic area and the sample size was limited, and the external validity of the findings may be constrained by geographic and population factors. Finally, the study could not provide a comprehensive understanding of the molecular mechanisms underlying the virus’s interaction with host cells and its impact on liver metabolism. While it offers macroscopic insights, such as changes in antibody levels and liver function indicators, it lacks detailed molecular-level analysis. These limitations highlight the need for further in-depth research.

Conclusions

In conclusion, our study showed a high SARS -CoV-2 infection rate but no severe cases among middle-aged and elderly individuals with hepatitis during the post-relaxation period of the “dynamic zero-COVID policy” in rural areas in northeast China. No significant impact of SARS-CoV-2 infection on the progression of hepatitis was observed, and four months post-infection, SARS-CoV-2-specific IgG and neutralizing antibody positivity remained high. Continuous monitoring of virus mutations remains crucial for public health, as new subvariants may evade immunity. Also, targeted prevention and care strategies for vulnerable populations can optimize resource allocation and protect their health.

Supplemental Information

Supplemental Information 1 Correlation of neutralizing antibody and IgG in HCC risk patients

(A) Total patients; (B) Uninfected; (C) Asymptomatic; (D) Symptomatic.

Supplemental Information 2 Methods of liver function tests and HCV RNA

Supplemental Information 3 Symptoms after infection in HCC high risk patients

Supplemental Information 4 Comparison of liver function at different time points and different infection states

Supplemental Information 5 Comparison of aMAP score at different time points and different infection states

Supplemental Information 6 Supplementary TableS5. Comparison of aMAP score of cirrhosis patients at different time points and different infection states

Supplemental Information 7 Long-covid 19 symptoms in HCC high risk patients

Supplemental Information 8 Comparison of neutralizing antibody and IgG in different groups

Supplemental Information 9 Univariate and multivariate logistic regression analyses of factors to long COVID-19

Supplemental Information 10 Univariate and multivariate logistic regression analyses of factors to reinfection

Supplemental Information 11 Raw data

The authors would like to thank all of those who participated in this study.

Additional Information and Declarations

Competing Interests

Author Contributions

Human Ethics

Data Availability

The authors declare there are no competing interests.

Yuchen Pan performed the experiments, analyzed the data, prepared figures and/or tables, authored or reviewed drafts of the article, and approved the final draft.

Zhifang Jia performed the experiments, analyzed the data, prepared figures and/or tables, authored or reviewed drafts of the article, and approved the final draft.

Xinyi Yu performed the experiments, prepared figures and/or tables, and approved the final draft.

Haiyong Lv performed the experiments, prepared figures and/or tables, and approved the final draft.

Yangyu Zhang performed the experiments, prepared figures and/or tables, and approved the final draft.

Yanhua Wu performed the experiments, authored or reviewed drafts of the article, and approved the final draft.

Jing Jiang conceived and designed the experiments, authored or reviewed drafts of the article, and approved the final draft.

The following information was supplied relating to ethical approvals (i.e., approving body and any reference numbers):

The study was approved by the Ethics Committee of the First Hospital of Jilin University (2023-455).

The following information was supplied regarding data availability:

The data are available in the Supplementary File.

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
