# Peer review of "Study on SARS-CoV-2 infection in middle-aged and elderly population infected with hepatitis virus: a cohort study in a rural area of northeast China"

_PeerJ, doi:10.7717/peerj.19021_

## Round 0.1 · original submission · Major Revisions

Authors need to refine their conclusions and discuss the limitations of their study. Please address all of the comments and provide point wise responses to both reviewers.

·

Basic reporting

The authors recruited 599 middle-aged people infected with HBV and HBC. Patients were given a questionnaire about their previous treatment for liver disease and their SARS2 infection status. Patients were tested for seropositivity rates and antibody levels associated with various symptoms. The incidence of reinfection was also monitored over a three-month period.
Their conclusion was that "although the rate of SARS-CoV-2 infection was high, there were no severe cases."
This conclusion is similar to that of numerous articles that have appeared since the beginning of covid until today, but there are also opposite conclusions. Many authors have described cohorts of patients with different levels of severity of liver disease during covid. These are essentially observational works that describe the disease status of their patients through symptoms and tests. The authors of this manuscript found no difference between infected and uninfected individuals in terms of liver enzyme tests. Of course, the lack of awareness of protective capabilities, as suggested by them, is a contributory cause that can favor infection at the individual level, but it cannot be used to explain that it is at the basis of Covid or the progression of patients towards cancer. The molecular mechanisms involved are metabolically very deep processes that have multiple multi-to-one and multi-multi relationships with the macroscopic observation level. Therefore, any macroscopic test has numerous apparent deep causes that must be studied individually to identify the real one. Added to this is that most individual genes/proteins have considerable redundancy and can take part in dozens of different molecular processes. This makes it very difficult to identify the real molecular causes.
When we say that the virus mutates, we refer to its ability to infect humans through Spike mutations, but we know very little, if not at a crude level, which molecular processes are activated by the virus's strategy and how these interact at the level of the individual patient's phenotype and metabolic homeostasis.
The study can be useful for health policies but it cannot give any motivational explanation on the disease because, whatever the aggressiveness of the virus, the outcome in terms of disease is always filtered by the infected human phenotype and its epigenetics that can totally change the fate of the viral progression. If the work reports only the observations of a small number of patients in a small area it is not very different from many others. If the authors then want to expand and frame the meaning of what they found in a general scientific context, then the work can show its validity.

Experimental design

The experimental design is valid.

Validity of the findings

Authors must frame their work in the general context of the infection. They describe specific metabolic events thus they should explain why they can or cannot provide answers at the various necessary levels (health, epidemiological, macroscopic metabolic, deep metabolic). In short, they must frame, explaining, what the meaning of their work is and what answers they can or cannot provide.

Additional comments

I suggest a broad reworking of the discussion in order to frame where their contribution falls within the general context of knowledge on covid.

Reviewer 2 ·

Basic reporting

Overall the article is interesting but needs many improvements in order to reach publication status. Apart from the issues found in experimental design and validity of findings sections, I think a few more issues need improvement.
1. In introduction section, in lines 59-61, it is stated that "SARS-CoV-2 infection rates and fatalities are higher in rural areas can be attributed to factors such as limited access to healthcare, poverty, and insufficient dissemination of health information". The reference for that should be added
2. In lines 66-68, more references regarding the worsening of pre-existing liver conditions should be added (ref: doi: 10.15403/jgld-5268; doi: 10.1016/j.jhep.2020.09.024)
3. The lines 270-274 in the discussion section should probably be in the introduction section instead, so that the discussion section becomes smaller, since the authors have written an extremely long discussion section. Likewise, the long covid definition should be moved in materials and methods section
4. The authors had a reinfection incidence of 17.5%. Were there any predisposing factors for that?

Experimental design

The experimental design of this manuscript is quite straightforward, but needs a few corrections/clarifications.
1. It is repeatedly written in the manuscript that patients included in this study where 'high risk in liver cancer'. Were the patients with liver cirrhosis, positive family history for HCC, high aMAP score to begin with or did they have any other reason to be considered as 'high risk' ? If that is not the case then this sentence should be deleted. Otherwise it should be clearly written in materials and methods section.
2. How were patients categorized as HCV (+) with just the antibody? HCV RNA should be done because many patients treated succesfully for HCV remain HCV Ab (+) (ref: doi:10.1093/cid/ciad319)
3. It should also be clearly stated when a patient was considered to be COVID-19 infected. IF a patient had SARS-CoV-2 infection a year ago was he considered to be COVID-19-infected t.ex.?
4. Moreover, the authors should state how a diagnosis of long covid (found in results section) was made in materials and methods section instead of the discussion section

Validity of the findings

The results section, provides much data, however I think many issues need to be re-evaluated.
1. The results regarding patients with liver cirrhosis are missing. Since these patients were considered as high-risk patients for severe COVID-19 (ref: doi: 10.1111/liv.14583; doi: 10.1053/j.gastro.2021.07.010) it would be important to see the outcomes in these patients
2. I am missing the multivariable results for the authors' findings. What were the predisposing factors for long COVID? Did, in a multivariable analysis, the level of antibodies predisposed to COVID-19 symptoms? Likewise for all your findings

---

## Round 0.2 · accepted · Accept

Authors have addressed all of the reviewers' comments and manuscript is ready for the publication.

·

Basic reporting

Manuscript well organized, with adeguate citations.

Experimental design

N.A. it is a review

Validity of the findings

Conlusion well stated.

Additional comments

I am the one who thanks the authors for taking my suggestions into consideration. The manuscript has been modified to make it fluent, very clear and understandable even to those who are not experts in the field.

Reviewer 2 ·

Basic reporting

The authors have substantially improved their manuscript, providing explanations in all my questions/remarks. English language has much improved and the new manuscript is professioanl enough and provides all available data to reach safe conclusions.

Experimental design

No comment

Validity of the findings

No comment

Additional comments

No comment